# Opioid prescribing in out-of-hours primary care in Flanders and the Netherlands: A retrospective cross-sectional study

Karin Hek[1]*, Tim Boogaerts[2], Robert A. Verheij[1,3], Hans De Loof[4], Liset van Dijk[1,5], Alexander L. N. van Nuijs[2], Willemijn M. Meijer[1], Hilde Philips[6]

1 Nivel, Netherlands Institute for Health Services Research, Utrecht, The Netherlands, 2 Toxicological Centre, University of Antwerp, Antwerp, Belgium, 3 Tranzo, Tilburg School of Social and Behavioral Sciences, Tilburg University, Tilburg, The Netherlands, 4 Laboratory of Physiopharmacology, University of Antwerp, Wilrijk, Belgium, 5 Groningen Research Institute of Pharmacy, Unit of PharmacoTherapy, Epidemiology & Economics, University of Groningen, Groningen, The Netherlands, 6 Unit of Family Medicine and Population Health, University of Antwerp, Gouverneur Kinsbergen Centrum, Wilrijk, Belgium

* k.hek@nivel.nl

**Data Availability Statement:** Access to data is subject to Nivel Primary Care Database and iCAREdata governance codes. Requests for access to the data can be directed at directie@nivel.nl

## Abstract

### Background

Increased opioid prescribing has raised concern, as the benefits of pain relief not always outweigh the risks. Acute and chronic pain is often treated in a primary care out-of-hours (OOH) setting. This setting may be a driver of opioid use but the extent to which opioids are prescribed OOH is unknown. We aimed to investigate weak and strong opioid prescribing at OOH primary care services (PCS) in Flanders (Northern, Dutch-speaking part of Belgium) and the Netherlands between 2015 and 2019.

### Methods

We performed a retrospective cross sectional study using data from routine electronic health records of OOH-PCSs in Flanders and the Netherlands (2015–2019). Our primary outcome was the opioid prescribing rate per 1000 OOH-contacts per year, in total and for strong (morphine, hydromorphone, oxycodone, oxycodone and naloxone, fentanyl, tapentadol, and buprenorphine and weak opioids (codeine combinations and tramadol and combinations) and type of opioids separately.

### Results

Opioids were prescriped in approximately 2.5% of OOH-contacts in both Flanders and the Netherlands. In Flanders, OOH opioid prescribing went from 2.4% in 2015 to 2.1% in 2017 and then increased to 2.3% in 2019. In the Netherlands, opioid prescribing increased from 1.9% of OOH-contacts in 2015 to 2.4% in 2017 and slightly decreased thereafter to 2.1% of OOH-contacts. In 2019, in Flanders, strong opioids were prescribed in 8% of the OOH-contacts with an opioid prescription. In the Netherlands a strong opioid was prescribed in 57% of these OOH-contacts. Two thirds of strong opioids prescriptions in Flanders OOH were issued for patients over 75, in the Netherlands one third was prescribed to this age group.

(Nivel-PCD) and icaredata@uantwerpen.be (iCAREdata). Restrictions involve establishing a data sharing agreement and approval by the appropriate Nivel Primary Care Database and iCAREdata governance bodies (privacy committee and steering committee). Data were anonymized and are accessible via https://doi.org/10.17026/dans-xuk-xb74 (DANS | Centre of expertise & repository for research data (knaw.nl), data from the Nivel Primare Care Database) and https://doi.org/10.6084/m9.figshare.19322813 (https://figshare.com/account/home, data from iCAREdata).

**Funding:** The research infrastructure "Nivel Primary Care Database" is funded by the Netherlands Ministry of Public Health, Welfare and Sports. The iCAREdata research dataplatform was funded by the Fund for scientific research of Flanders (FWO). The funders had no role in study design, data collection and analysis, decision to publish, or preparation of the manuscript.

**Competing interests:** AvN, HDL, HP, KH, RV, TB, WM have declared that no competing interests exist. LvD received funding for a project not related to this study from TEVA and Biogen. This does not alter our adherence to PLOS ONE policies on sharing data and materials.

## Conclusion

We observed large differences in strong opioid prescribing at OOH-PCSs between Flanders and the Netherlands that are likely to be caused by differences in accessibility of secondary care, and possibly existing opioid prescribing habits. Measures to ensure judicious and evidence-based opioid prescribing need to be tailored to the organisation of the healthcare system.

## Introduction

In Western countries opioid prescribing has increased rapidly over the past decades [1–3], mainly caused by an increase in opioid prescribing to treat chronic pain not related to cancer [4,5]. While short-term use of opioids is relatively safe, the use for chronic pain not related to cancer is problematic in the absence of evidence that the benefits outweigh the risk of developing opioid dependence, opioid use disorder, overdose and death [6–10].

The escalation in opioid prescribing in the United States of America (USA) and Canada, is accompanied by an increase in opioid-related hospitalization, mortality and number of patients in addiction care and has raised global concern [11]. In European countries, such as Belgium and the Netherlands, the use and misuse of prescription opioids also increased, but not to the same extent [3,12,13]. In Belgium, the number of prescription opioid sales increased with 6.8% between 2013 and 2019 with the highest increases reported for oxycodone and tramadol [14]. In the Netherlands the number of prescription opioid users nearly doubled between 2007 and 2017, from 4,100 users per 100,000 inhabitants to 7,500 per 100,000 inhabitants [12,15] with a four-fold increase in the number of oxycodone users.

In several European countries like the UK, Flanders (the Northern, Dutch speaking part of Belgium) and the Netherlands, out-of-hours primary care services (OOH-PCS) provide acute primary care during evenings, nights and weekends when the patient's own primary care provider is unavailable. Problems commonly presented at the primary care OOH setting include trauma, such as laceration and cuts, infections, such as upper respiratory infections or gastroenteritis, and abdominal pain [16]. Thus, like in other acute care settings, relief of acute pain and acute exacerbations of chronic pain [17] are essential activities of the primary OOH health care providers. Moreover, from previous studies in the acute care setting of emergency departments it is known that opioid prescribing in this acute setting may lead to persistent and high risk opioid use in up to 17% of patients starting with an opioid [18,19]. It is therefore important to investigate the extent of opioid prescribing in the primary care OOH setting.

In the current study we assess opioid prescribing in the primary care OOH setting in the neighbouring regions/countries, Flanders and the Netherlands. Both regions have concerns about the increasing national trends in opioid prescribing. However, the embedding of acute primary care in the health system differs. The scales of the OOH-PCSs in Flanders and the Netherlands are similar, but access to primary and secondary health care is organized differently. In Flanders, both the OOH-PCS and secondary (emergency) care are accessible without referral, whereas Dutch GPs (also in the OOH setting) have a gate keeper role for secondary and tertiary care, and patients are expected to contact the OOH-PCS by phone where triage is done prior to follow-up [16]. These differences in the organization of acute primary care may lead to differences in patient groups who consult the OOH-PCS [16] and thus also to differences in opioid prescribing.

The primary goal of this study is to study the prescribing of weak and strong opioids at the OOH-PCS between 2015 and 2019 in Flanders and the Netherlands. Additionally, this study explores which opioids were prescribed most frequently by GPs in OOH-PCSs in both regions.

## Methods

### Design and population

We performed a retrospective cross sectional study using routinely recorded electronic health records data of OOH-PCSs over the period 1 January 2015 to 31 December 2019 for Flanders and the Netherlands. In Flanders, data was acquired from iCAREdata [20,21], which comprises 5 to 9 OOH-PCSs over this five-year-period, covering more than 10% of Flanders' population. In the Netherlands, we used data collected in the Nivel Primary Care Database (Nivel-PCD, [22]). A database containing electronic health records data from 20 to 27 Dutch OOH-PCSs with a joint catchment area of more than half of the Dutch population (Table 1). The population in the catchment area of Nivel-PCD OOH-PCSs is representative for patient age and sex of the Dutch population. This is also the case in Flanders. Due to the extensive distribution over the whole of Flanders, a representative sample considering age and gender is available.

### Description of the data

We extracted patient age, sex, health problem (ICPC code, International Classification of Primary Care), contact date, contact type (face-to face consultation/home visit), and opioid prescriptions (ATC-code, Anatomic Therapeutic Classification) from the electronic health records of the OOH-PCS. We included the following opioids:

i.  strong opioids: morphine (N02AA01, N02AA51), hydromorphone (N02AA03), oxycodone (N02AA05), oxycodone and naloxone (N02AA55), fentanyl (N02AB03), tapentadol (N02AX06), buprenorphine (N02AE01)

ii.  weak opioids: codeine combinations (N02BE51, N02AA59, N02AJ06), tramadol (N02AX02), tramadol combinations (N02AJ13, N02AX52))

iii.  other opioids: nicomorphine (N02AA04), pethidine (N02AB02), dextromoramide (N02AC01), piritramide (N02AC03)

**Table 1. Description of the OOH-databases in Flanders and the Netherlands.**

|  | Flanders (iCARE database) | | | | | The Netherlands (Nivel Primary Care Database) | | | | |
|---|---|---|---|---|---|---|---|---|---|---|
|  | 2015 | 2016 | 2017 | 2018 | 2019 | 2015 | 2016 | 2017 | 2018 | 2019 |
| Number of OOH-PCSs# | 5 | 7 | 7 | 7 | 9 | 20 | 24 | 27 | 23 | 23 |
| Population number* | 803,809 | 1,124,449 | 1,131,191 | 1,124,046 | 1,604,329 | 8,236,133 | 10,757,942 | 11,259,938 | 9,057,552 | 10,549,980 |
| Number of contacts** | 56,350 | 80,492 | 81,898 | 88,672 | 130,621 | 2,108,919 | 2,703,643 | 2,770,026 | 2,308,766 | 2,656,910 |
| Number of contacts per 1000 inhabitants in catchment area | 70.1 | 71.6 | 72.4 | 78.9 | 81.4 | 256.1 | 251.3 | 246.0 | 254.9 | 251.8 |

# For NL: Number of OOH-PCS cooperations.

*FL has a total population of almost 7 million inhabitants; NL has a total population of approximately 17 million inhabitants.

**In Flanders, OOH care includes consultations and visits during weekends, in the Netherlands, OOH-care includes telephone consultations, consultations and visits during working day evenings, and weekends (numbers for the Netherlands when excluding contacts during evenings and nights on working days and telephone contacts: 806,919, 1,022,721, 1,023,728, 815,414, 906,621).

To improve the comparability of the Flemish and Dutch dataset, we performed an additional analysis in which we excluded the following contacts: (i) telephone contacts in the Netherlands as these are not available in the Flemish system, (ii) contacts on weekdays in the Netherlands. Dutch OOH-PCSs are also open on evenings and nights of weekdays, while Flemish OOH-PCS are not. Therefore, in this additional analysis, for both regions, we only used data from the weekends (19 p.m. on Fridays until 7 a.m. on Mondays).

### Data analysis

To assessopioid prescribing over 2015–2019 at the OOH-PCS, we calculated the number of OOH-PCS contacts during which an opioid was prescribed per 1000 OOH-PCS contacts per year. We did this for all opioids combined, and for strong and weak opioids separately. Contacts in which both types of opioids were prescribed were included in both groups. We also assessed the distribution of specific types of opioids, such as morphine and oxycodone, in Flemish and Dutch OOH-PCSs per year. We determined the most common diagnoses for which an opioid was prescribed. Contacts with a missing ICPC code were included in the analyses (0% for Flanders and 5% for the Netherlands).

### Ethical approval

Flanders: iCAREdata received approvals concerning patients participation and opt-out options from the Sectoral Committee of Social Security and Health of the Privacy Commission (Beraadslaging_AG_094_2014 and Beraadslaging_AG_094_2014bis) and from the Ethics Committee of the Antwerp Academic Hospital (Approval 13/34/330, dd 02/09/2013). Data were pseudonymized and did not comprise any directly identifying personal information such as names, addresses and citizen service number.

The Netherlands: The use of personal data for research purposes in the Netherlands is regulated under the Dutch Medical Treatment Contracts Act (WGBO). The WGBO stipulates that explicit consent is not required if a) requesting consent is not reasonably possible (if for example the patient is deceased) or- if b) the request for permission cannot reasonably be expected from the caregiver. The latter can refer to situations in which too great effort an effort is needed from health care providers to, or when asking for permission would lead to a selective response. However, data collection should take place taking into account all possible organizational and technical measures needed. In addition the Medical Research Involving Human Subjects Act (WMO), stipulates that approval by one of the national medical ethical committees is required only if the research involves humans subjected to actions or if rules of behavior are imposed on them. This is not the case in our study. OOH-PCSs that participate in Nivel-PCD are contractually obliged to: (i) inform their patients about their participation in Nivel-PCD and (ii) to inform patients about the option to opt-out for inclusion of their data in the database. Data were pseudonymized before leaving the health care organization's premises and did not comprise any directly identifying personal information such as names, addresses and citizen service number [23]. Neither obtaining informed consent from patients nor approval by a medical ethics committee is obligatory for observational studies containing no directly identifiable data (Dutch Civil Law, Article 7: 458). The study was approved according to the governance code of Nivel-PCD under number: NZR-00319.034, and all legally required technical and organizational measures were applied to avoid real life identification of subjects.

### Results

Between 2015 and 2019, approximately 1.9 to 2.5% of OOH contacts resulted in an opioid prescription. Fig 1 and S1A (the Netherlands) and S1B (Flanders) Table show the changes in total,

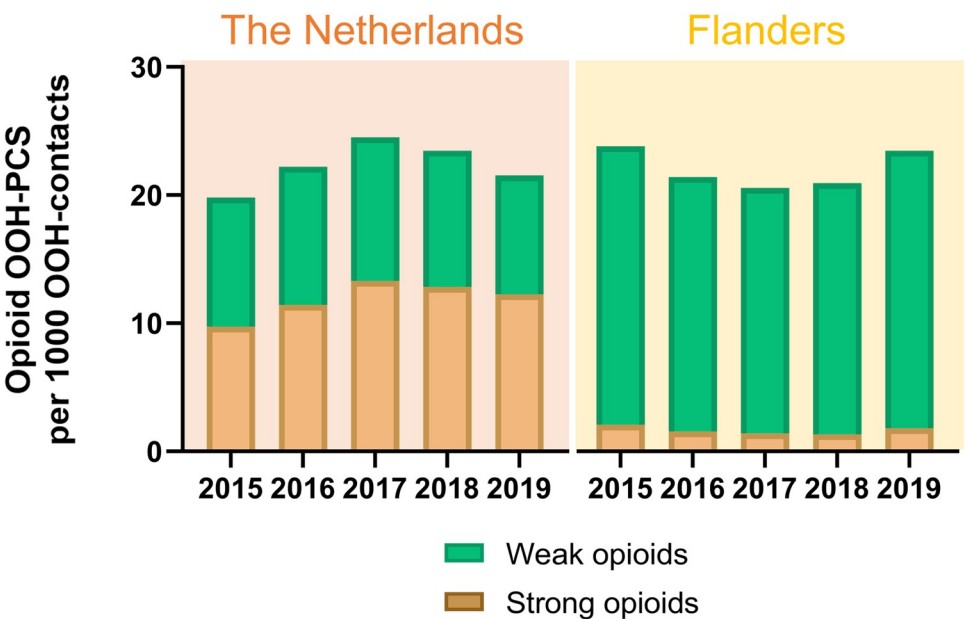

**Fig 1. Opioid prescribing in out-of-hours primary care in the Netherlands (upper panel) and Flanders (lower panel) between 2015 and 2019, number of contacts with an opioid prescription per 1000 OOH-contacts.** OOH-PCS = out-of-hours primary care service.

weak and strong opioid prescribing over time. In Flanders, the number of contacts with an opioid prescription decreased from 24.1 per 1000 OOH contacts in 2015 to 20.6 per 1000 OOH contacts in 2017 (relative decrease of 14.5%) and increased thereafter to 23.5 per 1000 OOH contacts in 2019. In the Netherlands, opioid prescribing increased from 19.8 per 1000 OOH contacts in 2015 to 24.5 per 1000 OOH contacts in 2017 (relative increase of 23.7%), and decreased thereafter to 21.4 per 1000 OOH contacts in 2019. S1B Table shows the same numbers for the Netherlands when excluding weekday and telephone contacts, comparable to OOH-care in Flanders. The relative changes over the years are similar, but the opioid prescribing rate is somewhat higher (ranging between 21.7 and 28.0).

Prescription rates for strong opioids in OOH care decreased from 2.1 per 1000 contacts in 2015 to 1.3 in 2018 and increased to 1.8 per 1000 contacts in 2019 (14.3% relative decrease between 2015 and 2019). In the Netherlands, strong opioid prescribing increased from 9.7 to 13.3 per 1000 contacts between 2015 and 2017 (37.1% increase) and slightly decreased thereafter to 12.3 per 1000 contacts in 2019.

## Type of opioid prescribed

Fig 2 and S1A Table show the specific types of opioids prescribed in OOH-PCSs in Flanders and the Netherlands. In Flanders, fentanyl, morphine and oxycodone were the most frequently prescribed strong opioids (in 2019 0.3, 0.9 and 0.6 per 1000 contacts respectively). In the Netherlands, morphine and oxycodone were the most commonly prescribed strong opioids over time (in 2019 6.9 and 4.5 per 1000 contacts). In Flanders, tramadol is the most frequently prescribed weak opioid (11.4 per 1000 contacts in 2019), but the prescribing of tramadol combinations and codeine combinations is also substantial, as illustrated by Fig 2 (4.8 and 3.7 per 1000 contacts in 2019). Tramadol was the most frequently prescribed weak opioid in the Netherlands (approximately 80% of all weak opioid prescriptions, 7.8 per 1000 contacts in 2019).

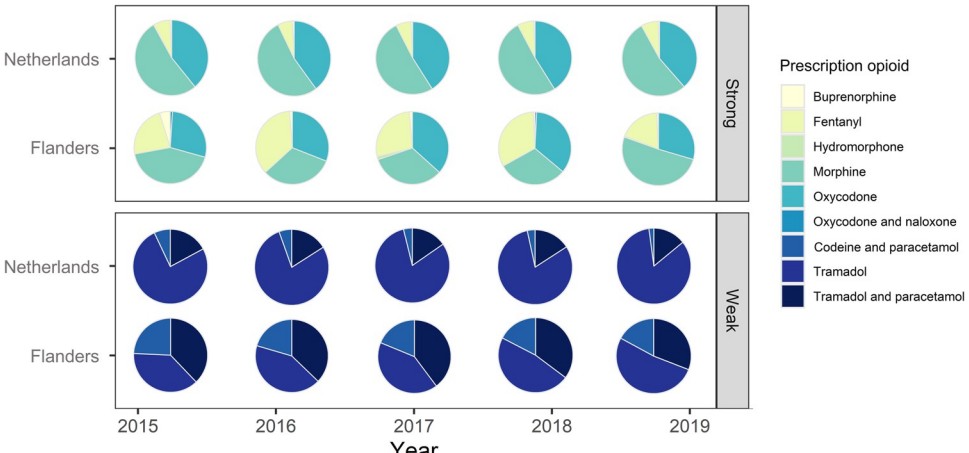

**Fig 2. Distribution of type of opioid prescribed in out-of-hours primary care, separately for strong (upper panel) and weak opioids (lower panel) in Flanders and the Netherlands.** The figure does not reflect the relative proportions in prescribing of weak and strong opioids over time (see Fig 1).

## Diagnoses for which opioids were prescribed

There was high concordance in both countries on the top-10 diagnoses recorded with a prescription for weak opioids (S2 Table). This list included mostly back-pain and other locomotor system pain, as well as, at lower frequencies, dental pain and stomach pain. These also topped the list of diagnoses recorded with strong opioids, along with dyspnea. In the Netherlands bile duct problems and kidney stones also ranked in the top 10 of diagnoses recorded with strong opioids.

## Characteristics of patients with an opioid prescription

Table 2 shows the patient characteristics of patients with an OOH opioid prescription. More than half the patients with an opioid (either strong or weak) was female, both in Flanders and the Netherlands. The age distribution of patients with weak opioids was comparable between Flanders and the Netherlands. However, the age distribution of strong opioid users differs between the countries. Strong opioids were mainly prescribed to patients aged 75 or older in Flanders (64% vs 30% in the Netherlands). The sex and age distribution of patients with an opioid prescription OOH did not change during the study period (S3 Table).

**Table 2. Age and sex distribution of patients with an opioid prescription at the out-of-hours service, in Flanders and the Netherlands in 2019.**

|  | All opioids | | Weak opioids | | Strong opioids | |
|---|---|---|---|---|---|---|
|  | **FL** | **NL** | **FL** | **NL** | **FL** | **NL** |
| **Total** | | | | | | |
| Sex (% females) | 58.2 | 56.4 | 58.1 | 58.2 | 60.0 | 55.2 |
| Age category (%) | | | | | | |
| 0–14 |  | 0.1 | 0.2 | 0.2 | 0 | 0.1 |
| 15–24 | 0.2 | 5.3 | 6.5 | 7.8 | 0 | 3.3 |
| 25–44 | 6.0 | 26.0 | 34.9 | 32.8 | 7.0 | 20.6 |
| 45–64 | 32.7 | 32.4 | 35.2 | 34.5 | 13.5 | 30.6 |
| 65–74 | 33.6 | 13.7 | 9.5 | 11.7 | 15.2 | 15.3 |
| 75+ | 9.9 | 22.4 | 13.7 | 13.1 | 64.3 | 30.1 |
|  | 17.6 | | | | | |

## Discussion

In this study, we analysed the weak and strong opioid prescribing in OOH-PCSs in both Flanders and the Netherlands between the years 2015 and 2019. In Flanders we observed a slight decrease in OOH opioid prescribing between 2015 and 2019, with the lowest prescription rates in 2017. This trend was present for both weak and strong opioids. For the Netherlands, we observed an increase in OOH opioid prescribing between 2015 and 2017, and a decrease in 2018 and 2019. The increase between 2015 and 2017 was driven by increased prescribing of strong opioids, while weak opioids prescribing remained stable. These changes (both the increase and the decrease thereafter) are consistent with the opioid prescribing trends in the Netherlands in the non-acute primary care setting [15]. The observed decrease in opioid prescribing is likely the result of increased attention since the end of 2017 for the steep rise in and risks of opioid prescribing. In the Netherlands, more than half of all opioids prescribed at the OOH-PCS were strong opioids, which was also comparable to figures in non-acute primary care [15,24].

### Prescribing of strong opioids

We observed a marked difference in prescribing of weak and strong opioids between Flanders and the Netherlands. This may be explained by differences in the OOH-setting resulting in a different patient population. This suggests that OOH-PCSs in the Netherlands, which have a gate keeping role, are consulted for more severe cases than OOH-PCSs in Flanders, where secondary care, but also the OOH-PCS, is directly accessible. Furthermore, in the Netherlands patients are expected to contact the OOH-PCS by phone, when formal triage is done, after which follow-up is done on telephone, at a consultation, or in a home visit. This is corroborated by the difference in diagnoses that opioids are prescribed for during OOH. While kidney stones were the main reason for prescribing strong opioids in OOH-PCS in the Netherlands, this diagnosis was not among the top-10 diagnoses with strong opioids prescribed in OOH care in Flanders. It seems likely that patients suffering from kidney stones directly consulted the hospital emergency department in Flanders.

### Type of opioid prescribed

In Flanders tramadol/paracetamol combinations were often prescribed, in spite of the weak scientific evidence for their use [25,26] and their absence from the guidelines [27,28]. This may be related to the reimbursement status of paracetamol in Belgium. Paracetamol-only products are only reimbursed in Belgium for chronic use after additional administrative hurdles that prelude its use in acute pain [14]. A similar situation holds for the codeine/paracetamol combination products, which could explain their lower share in the OOH-PCSs in Flanders. In marked contrast, tramadol/paracetamol combinations are routinely reimbursed and frequently prescribed. Furthermore, substantial amounts of fentanyl were prescribed in Flemish OOH-PCSs. These are not indicated for acute pain relief or as first choice opioid in case a strong opioid is indicated. However, it is difficult to assess the validity of these OOH prescriptions as they may be an extension of an existing treatment with transdermal fentanyl. In that case it however remains questionable whether these people require OOH-care to obtain a prescription for a chronic condition.

In the Dutch OOH-PCS setting morphine was the most commonly prescribed strong opioid, followed by oxycodone. This differed from trends in opioid prescribing in general practices, where oxycodone is the most commonly prescribed strong opioid [12,15]. During OOH Dutch GPs hardly prescribe fentanyl, whereas in non-acute general practice, fentanyl was prescribed as often as morphine [12,15]. This difference in type of opioid may be explained by a

difference in health problems that are encountered during OOH compared to non-acute general practice. Kidney stones were the most common diagnosis for which Dutch GPs prescribed a strong opioid during OOH. To quickly relieve acute severe pain, patients with kidney stones are injected with morphine.

### Opioid prescribing for locomotor system problems

Both in Flanders and in the Netherlands, strong opioids were prescribed for back pain and related locomotor system problems. Strong opioids are not intended for chronic use in these type of health problems [29]. However, from the available data, we could not deduct the amount or dosage of prescribed opioids and whether they were initiated at the OOH-PCS or were already used chronically.

### Age distribution of patients with an opioid

In Flanders, strong opioids were mainly prescribed to older adults, while in the Netherlands, these were also often prescribed to middle-aged adults. This may be explained by the difference in access to secondary care and presented health problems at the OOH-PCS in the two regions [16]. In general, it is advised to be cautious when prescribing weak opioids to older adults, as it may cause mental confusion [30]. Use of strong opioids leads to an increased fall risk, which is related to mortality in older adults [31].

### Study strengths and limitations

The strength of this study is that we rely on large datasets derived from routinely recorded electronic health records to assess opioids prescribing in OOH primary care over a period of five years in two adjacent but different regions/countries, Flanders and the Netherlands. This study, however, also has a number of limitations. First, we did not have precise information on the amount and duration of opioids that were prescribed. Both amount and duration are related to risks for patients [32] and are therefore important indicators of the quality of opioid prescribing. Second, for analysis of the indications for opioid prescription we depended on diagnosis recording by the GP. Third, we did not assess whether opioid prescribing was the start of chronic and potentially problematic opioid use. Last, we did not statistically test whether changes in opioid prescribing over time were significant.

### Implications of the study findings

In approximately 2 to 2.5% of all OOH-PCS contacts an opioid was prescribed in both Flanders and the Netherlands. There are no signs of a strong increase in opioid prescribing at OOH-PCSs in Flanders and the Netherlands over the years. Nevertheless, we did observe a marked difference between Flanders and the Netherlands in the prescribing of strong opioids, that are likely caused by differences in the accessibility of secondary care. This implies that the development of measures to stimulate appropriate (strong) opioid prescribing should take into account the organisation of the health care system. Further study on the appropriateness of these prescriptions is needed to determine whether opioid prescriptions in the OOH-PCS are a driver of problematic opioid use.

## Conclusion

In this first database study of out-of-hours primary care opioid prescribing we did not observe a large increase in OOH-PCS opioid prescribing between 2015 and 2019 in Flanders, nor in the Netherlands. We did observe large differences in strong opioid prescribing between the

two neighbouring regions that are likely to be caused by differences in accessibility of secondary care, and possibly existing opioid prescribing habits. Measures to control opioid prescribing should thus be developed taking into account the organisation of the health care system.

## Supporting information

**S1 Table. Number of contacts with at least one opioid prescription at the OOH-PCS per 1000 OOH-PCS contacts in 2015–2019 in the Netherlands and in the weekend at the OOH-PCS per 1000 OOH-PCS weekend-contacts in 2015–2019 in Flanders and the Netherlands.**
(DOCX)

**S2 Table. Top 10 diagnoses of OOH contacts with $\geq$ 1 opioid prescription in Flanders and the Netherlands in 2018.**
(DOCX)

**S3 Table. Sex and age distribution of patients with an opioid prescription at the OOH-PCS, in Flanders and the Netherlands in 2015–2019.**
(DOCX)

## Author Contributions

**Conceptualization:** Karin Hek, Tim Boogaerts, Robert A. Verheij, Hans De Loof, Liset van Dijk, Alexander L. N. van Nuijs, Willemijn M. Meijer, Hilde Philips.

**Formal analysis:** Karin Hek, Tim Boogaerts, Willemijn M. Meijer.

**Methodology:** Karin Hek, Tim Boogaerts, Willemijn M. Meijer.

**Supervision:** Robert A. Verheij, Liset van Dijk, Alexander L. N. van Nuijs, Hilde Philips.

**Validation:** Karin Hek, Tim Boogaerts, Willemijn M. Meijer.

**Visualization:** Tim Boogaerts.

**Writing – original draft:** Karin Hek, Tim Boogaerts, Hans De Loof, Willemijn M. Meijer.

**Writing – review & editing:** Karin Hek, Tim Boogaerts, Robert A. Verheij, Hans De Loof, Liset van Dijk, Alexander L. N. van Nuijs, Willemijn M. Meijer, Hilde Philips.

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
