## [Decision Letter · Decision Letter 0]

2 Dec 2021

PONE-D-21-33076Opioid prescribing in out-of-hours primary care in Flanders and the Netherlands: a retrospective observational studyPLOS ONE

Dear Dr. Hek,

Thank you for submitting your manuscript to PLOS ONE. After careful consideration, we feel that it has merit but does not fully meet PLOS ONE’s publication criteria as it currently stands. Therefore, we invite you to submit a revised version of the manuscript that addresses the points raised during the review process.

We look forward to receiving your revised manuscript.

Kind regards,

Vijayaprakash Suppiah, PhD

Academic Editor

PLOS ONE

Journal Requirements:

2. You indicated that ethical approval was not necessary for your observational study conducted within the Netherlands. We understand that the framework for ethical oversight requirements for studies of this type may differ depending on the setting and we would appreciate some further clarification regarding your research. Could you please provide further details on why your study is exempt from the need for approval and confirmation from your institutional review board or research ethics committee (e.g., in the form of a letter or email correspondence) that ethics review was not necessary for this study? Please include a copy of the correspondence as an ""Other"" file.

3. In ethics statement in the manuscript and in the online submission form, please provide additional information about the patient records/samples used in your retrospective study. Specifically, please ensure that you have discussed whether all data/samples were fully anonymized before you accessed them and/or whether the IRB or ethics committee waived the requirement for informed consent. If patients provided informed written consent to have data/samples from their medical records used in research, please include this information.

[AvN, HDL, HP, KH, RV, TB, WM have declared that no competing interests exist. 

LvD received funding for a project not related to this study from TEVA.] 

Reviewers' comments:

Reviewer's Responses to Questions

**Comments to the Author**

1. Is the manuscript technically sound, and do the data support the conclusions?

Reviewer #1: Yes

Reviewer #2: Yes

2. Has the statistical analysis been performed appropriately and rigorously? 

Reviewer #1: Yes

Reviewer #2: Yes

3. Have the authors made all data underlying the findings in their manuscript fully available?

Reviewer #1: Yes

Reviewer #2: Yes

4. Is the manuscript presented in an intelligible fashion and written in standard English?

Reviewer #1: Yes

Reviewer #2: Yes

5. Review Comments to the Author

Reviewer #1: The authors present a retrospective observational study examining data from routine electronic health records (EMRs) of patients in the Flanders and the Netherlands from 2015 to 2019. They analyzed the number of contacts in which GPs prescribed an opioid and the type of opioids that were prescribed. The opioid crisis in the United States has increased concerns about appropriate opioid prescribing and the significance of clearly defining the benefits and risks of utilizing opioid analgesics in patients suffering from pain. Evaluating real-world prescribing patterns can provide better insight into the creation of national guidelines.

The manuscript is engaging, timely, impactful, and interesting. The described manuscript format is appropriate, and there are ethical issues noted. The manuscript is referenced and structured correctly, and the results support the conclusions. The authors described the study's limitations which might question the generalizability of the data and therefore should suggest that further studies are needed.

Reviewer #2: Thank you for the opportunity to review this retrospective cross-sectional study that examines out-of-hours primary care opioid prescribing in the Netherlands and Flanders. The paper examines an often overlooked source of opioid prescriptions and utilizes two data sources that provide sufficient generalizability for their study areas. The scope is somewhat limited to health systems and countries that have similar models, but that is true of many studies and is hardly a criticism. I think the strength of the study is in their comprehensive data source and straight forward design. I do think the attempts to compare the two regions came up short; there weren't any statistical analyses comparing the two regions. However the authors do an admirable job of explaining why the regions differ in the rates/types of opioid prescribed, interpreting the results within the context of the local environment, and provide the non-European reader (Me) an understanding of the healthcare environment in those two countries/regions. I strongly question the decision to remove the weekday visits from the Netherlands analysis. I can appreciated that this was done to provide a more direct comparison between the two regions. But since there were no analyses done, you've just removed a large number of contacts (visits?) (2 million per year!!!!) and severely biases your study. I would think that by including these visits the reader would gain a much more complete understanding of the role of OOH-PCSs in opioid prescribing, which I believe to be the true benefit of the study (not the regional comparison). In the event that you want to statistically compare the regions, these can be removed.

Specific Comments:

Abstract

Background, Page 2, line 37-38: delete "that differ in the way..." as it is very confusing to read and is not necessary for the abstract

Methods: need to provide more info- study is a retrospective cross sectional study; what is the primary outcome (mean yearly rate of opioid prescriptions per 1000 visits); what are strong v weak opioids; define GP. Also, if you aren't doing a statistical analyses of the yearly rate, you shouldn't use the term "trend". I think it would be very easy to do a piece-wise linear regression using 2017 as a midpoint.

Results, Page 2, line 44-48: Would rewrite the description of the opioid prescribing rates in Flanders to read more like the Netherlands description (rates went from xx to y between 2015 - 2017 and then increased to zzz in 2019.

Page 2, line 48-49: "...OOH-contacts concerning an opioid prescription". I'm assuming that you mean that strong opioids were prescribed 8% of the time WHERE an opioid was prescribed. I would just clarify as it is a little confusing as written.

Conclusion, Page 2, line 55-56: I would rewrite this to something of the effect of "Measures to ensure judicious and evidenced-based prescribing of opioids need to be tailored to the local healthcare system organization".

Introduction

Page 3, line 64: "...benefits outweigh the risk of developing opioid dependence, opioid use disorder, overdose..."

Page 3, line 74-75: "Acute and chronic pain..." this line is out of place in context with the preceding paragraph

Page 3, line 76: I would provide the "definition" of Flanders earlier in the manuscript

Page 3, line 82-84: I would provide an actual number describing the risk of persistent opioid use after prescription (could use Shah A, Hayes CJ, Martin BC. Characteristics of Initial Prescription Episodes and Likelihood of Long-Term Opioid Use - United States, 2006-2015. MMWR Morb Mortal Wkly Rep. 2017;66(10):265-269)

Page 3, line 91: "...with each 80 to 160 GPs" Is this the number of GPs per OOH?

Page 3-4, line 90-106: This explanation of the differences in the care models of the Netherlands and Flanders is excellent and completely necessary to interpret the findings of the study. However it needs to be condensed and moved to the discussion. You can introduce a very brief explanation in the intro to justify why you are comparing the two regions, but the bulk of the description should be reserved for the discussion

Methods

Page 4, line 113: I would think this is a retrospective cross sectional study

Page 4, line 116 and line 121: Does iCAREdata cover 10% or 66% of the Flemish population?

Page 6, line 146-148: I think it is unwise to exclude these data. The comparison between the regions seems like a secondary aim. You are excluding 2 million contacts per year...that's a lot. I would strongly encourage you to include these visits in your yearly descriptions and in the event you go back and perform a statistical analyses comparing the regions, you can then remove them. Trying to describe this very narrow subset of weekend contacts that result in an prescription is just not nearly as helpful.

Page 6, line 150: ALL means need to have 95% CI

Results

All values that represent mean need 95% CI. And please be consistent with the number of significant figures you are using.

Page 6, line 171-172: Would re-write first sentence "Between 2015-2019, XXX (2.5%) OOH-contacts resulted in an opioid prescription (Figure 1, Table S1)"

Page 6-7, line 173-177: I am not following your % decrease calculation; going from 24  21% is an absolute reduction of 3%. 1-(21/24) = 12.5% relative reduction. If my math is incorrect please let me know

Page 7, line 178-179: I would continue to present the mean (with 95% CI) values instead of switching to range.

Page 7, line 188: Give % for each opioid

Discussion

I would encourage you to include some discussion on how these results compare to other studies that describe opioid prescribing, particularly those that examine sources of opioid prescribing. Listed are references that I am familiar with as a US-based provider (Temporal Trends in Opioid Prescribing Practices in Children, Adolescents, and Younger Adults in the US From 2006 to 2018, JAMA Pediatr

. 2021 Oct 1;175(10):1043-1052. doi: 10.1001/jamapediatrics.2021.1832. Opioid Prescribing to US Children and Young Adults in 2019, Pediatrics

. 2021 Sep;148(3):e2021051539. doi: 10.1542/peds.2021-051539. Epub 2021 Aug 16. Variation in Adult Outpatient Opioid Prescription Dispensing by Age and Sex — United States, 2008–2018, MMWR Morb Mortal Wkly Rep. 2020 Mar 20;69(11):298-302. doi: 10.15585/mmwr.mm6911a5.).

Page 8, line 217-218: "Overall we did...nor in the Netherlands." Were you expecting to see a large increase? Was that your hypothesis? Seems odd to state this here, would delete.

Page 8, line 223-225: Are you saying that the overall trend of decreasing opioid prescribing is secondary to the increased knowledge of the dangers of opioids?

It is confusing as this sentence just follows your statement that prescribing of strong opioids increased between 2015-2019 so it makes it seem like you are saying the attention being paid to prescribing has lead to increased strong opioid prescribing

Page 9, line 254: Please provide a reference

Page 9-10: Given the high prevalence of tramadol in your sample, I would include some discussion on the recent paper that showed increased mortality with tramadol (Association of Tramadol vs Codeine Prescription Dispensation With Mortality and Other Adverse Clinical Outcomes, JAMA. 2021 Oct 19;326(15):1504-1515. doi: 10.1001/jama.2021.15255.)

Limitations: IF you are going to continue to exclude the weekday contacts in the Netherlands, it needs to be listed in the limitations as this is excluding the majority of contacts from the Netherlands and significantly biases the results. As stated previously, would strongly encourage you to include these results and perform trend analysis.

Tables and Figures

Try to use "." or "," as decimal points...you go back and forth in the supplemental material

Figure 1: Year should be on the X-axis

6. PLOS authors have the option to publish the peer review history of their article (what does this mean?). If published, this will include your full peer review and any attached files.

Reviewer #1: **Yes: **Joseph V Pergolizzi, MD

Reviewer #2: No

---

## [Author Response · Author response to Decision Letter 0]

4 Feb 2022

We thank the reviewers for their valuable comments that helped us to improve our manuscript. We provide a point-by-point reply below. Numbering of the comments starts at 7, as comment 1 to 6 were comments regarding journal requirements, that were replied to separately. 

Reviewer #1: 

7. 

The authors present a retrospective observational study examining data from routine electronic health records (EMRs) of patients in the Flanders and the Netherlands from 2015 to 2019. They analyzed the number of contacts in which GPs prescribed an opioid and the type of opioids that were prescribed. The opioid crisis in the United States has increased concerns about appropriate opioid prescribing and the significance of clearly defining the benefits and risks of utilizing opioid analgesics in patients suffering from pain. Evaluating real-world prescribing patterns can provide better insight into the creation of national guidelines.

The manuscript is engaging, timely, impactful, and interesting. The described manuscript format is appropriate, and there are ethical issues noted. The manuscript is referenced and structured correctly, and the results support the conclusions. The authors described the study's limitations which might question the generalizability of the data and therefore should suggest that further studies are needed.

Author’s reply: We thank the reviewer for the compliments. We agree with the reviewer that further study is required to shed further light on the appropriateness of opioid prescriptions out-of-hours and whether this is a driver of problematic opioid use. We included a sentence on required further study in the implications section in the discussion of the manuscript: “Further study on the appropriateness of these prescriptions is needed to determine whether opioid prescriptions in the OOH-PCS are a driver of problematic opioid use.”

Reviewer #2: 

8. 

Thank you for the opportunity to review this retrospective cross-sectional study that examines out-of-hours primary care opioid prescribing in the Netherlands and Flanders. The paper examines an often overlooked source of opioid prescriptions and utilizes two data sources that provide sufficient generalizability for their study areas. The scope is somewhat limited to health systems and countries that have similar models, but that is true of many studies and is hardly a criticism. I think the strength of the study is in their comprehensive data source and straight forward design. I do think the attempts to compare the two regions came up short; there weren't any statistical analyses comparing the two regions. However the authors do an admirable job of explaining why the regions differ in the rates/types of opioid prescribed, interpreting the results within the context of the local environment, and provide the non-European reader (Me) an understanding of the healthcare environment in those two countries/regions. I strongly question the decision to remove the weekday visits from the Netherlands analysis. I can appreciated that this was done to provide a more direct comparison between the two regions. But since there were no analyses done, you've just removed a large number of contacts (visits?) (2 million per year!!!!) and severely biases your study. I would think that by including these visits the reader would gain a much more complete understanding of the role of OOH-PCSs in opioid prescribing, which I believe to be the true benefit of the study (not the regional comparison). In the event that you want to statistically compare the regions, these can be removed.

Author’s reply: we thank the reviewer for the elaborate review and followed the suggestion of the reviewer to provide a complete understanding of the role of OOH-PCSs in opioid prescribing in the Netherlands, by including all contacts. For Flanders it was not necessary to distinct between week and weekend-contacts since the OOH-PCS services are only available for primary care during the weekends and national holidays. We reply to specific comments below. 

Specific Comments:

Abstract

9. 

Background, Page 2, line 37-38: delete "that differ in the way..." as it is very confusing to read and is not necessary for the abstract

Author’s reply: we removed this from the abstract as suggested. 

10. 

Methods: need to provide more info- study is a retrospective cross sectional study; what is the primary outcome (mean yearly rate of opioid prescriptions per 1000 visits); what are strong v weak opioids; define GP. 

Author’s reply: We indeed used a retrospective cross sectional design and updated this term throughout the manuscript. Furthermore, we updated our methods section (i.e. definitions of strong and weak opioids, an definition of GP) in the abstract according to the reviewer’s suggestions. 

11.

Also, if you aren't doing a statistical analyses of the yearly rate, you shouldn't use the term "trend". I think it would be very easy to do a piece-wise linear regression using 2017 as a midpoint.

Author’s reply: As we have only 5 data points (5 years analyzed), we chose not to do a piece wise linear regression analysis. We refer to this in the limitation section of the discussion: “Last, we did not statistically test whether changes in opioid prescribing over time were significant.”. Although we did not statistically test changes over time, we do provide numbers that provide insight in changes in opioid prescribing over time. We believe therefore, we can still talk about trends in opioid prescribing. We adjusted the manuscript to avoid terminology that suggests that we statistically test trends and also avoid the word “trend”. 

12. 

Results, Page 2, line 44-48: Would rewrite the description of the opioid prescribing rates in Flanders to read more like the Netherlands description (rates went from xx to y between 2015 - 2017 and then increased to zzz in 2019.

Author’s reply: This was changed as suggested into: “In Flanders, OOH opioid prescribing went from 2.4% in 2015 to 2.1% in 2017 and then increased to 2.3% in 2019.”

13. 

Page 2, line 48-49: "...OOH-contacts concerning an opioid prescription". I'm assuming that you mean that strong opioids were prescribed 8% of the time WHERE an opioid was prescribed. I would just clarify as it is a little confusing as written.

Author’s reply: Indeed, that is what we meant. We adjusted as follows: “In 2019, in Flanders, strong opioids were prescribed in 8% of the OOH-contacts with an opioid prescription.”

14. 

Conclusion, Page 2, line 55-56: I would rewrite this to something of the effect of "Measures to ensure judicious and evidenced-based prescribing of opioids need to be tailored to the local healthcare system organization".

Author’s reply: we changed the conclusion of the abstract as suggested into: “Measures to ensure judicious and evidence-based opioid prescribing need to be tailored to the organisation of the healthcare system.”

Introduction

15. 

Page 3, line 64: "...benefits outweigh the risk of developing opioid dependence, opioid use disorder, overdose..."

Author’s reply: we changed this as suggested into: “the benefits outweigh the risk of developing opioid dependence, opioid use disorder, overdose and death.”

16. 

Page 3, line 74-75: "Acute and chronic pain..." this line is out of place in context with the preceding paragraph

Author’s reply: we removed this sentence from the manuscript. As in the following paragraph we already mention the following: “Thus, like in other acute care settings, relief of acute pain and acute exacerbations of chronic pain are essential activities of the primary OOH health care providers.”

17. 

Page 3, line 76: I would provide the "definition" of Flanders earlier in the manuscript

Author’s reply: we left the definition of Flanders at the same position in the text, as it is the first time that Flanders is mentioned in the manuscript. The numbers mentioned in the paragraph before refer to Belgium, not to Flanders only. We added the definition of Flanders to the abstract. 

18. 

Page 3, line 82-84: I would provide an actual number describing the risk of persistent opioid use after prescription (could use Shah A, Hayes CJ, Martin BC. Characteristics of Initial Prescription Episodes and Likelihood of Long-Term Opioid Use - United States, 2006-2015. MMWR Morb Mortal Wkly Rep. 2017;66(10):265-269)

Author’s reply: thank you for providing this reference. We did not include the reference in the paragraph as it does not refer to long-term opioid use starting in acute care specifically. Therefore, we chose to include an estimate from studies in the acute care setting. 

This sentence now reads as follows: “Moreover, from previous studies in the acute care setting of emergency departments it is known that opioid prescribing in this acute setting may lead to persistent and high risk opioid use in up to 17% of patients starting with an opioid.”

19. 

Page 3, line 91: "...with each 80 to 160 GPs" Is this the number of GPs per OOH?

Author’s reply: This was indeed the number of GPs per OOH. However, based on the following comment of the reviewer, we removed this part from the introduction and added an adjusted paragraph to the discussion. (see comment 20)

20. 

Page 3-4, line 90-106: This explanation of the differences in the care models of the Netherlands and Flanders is excellent and completely necessary to interpret the findings of the study. However it needs to be condensed and moved to the discussion. You can introduce a very brief explanation in the intro to justify why you are comparing the two regions, but the bulk of the description should be reserved for the discussion

Author’s reply: we condensed the paragraph in the introduction and added parts of the original paragraph to the discussion. 

The introduction now reads as follows: “The scales of the OOH-PCSs in Flanders and the Netherlands are similar, but access to primary and secondary health care is organized differently. In Flanders, both the OOH-PCS and secondary (emergency) care are accessible without referral, whereas Dutch GPs (also in the OOH setting) have a gate keeper role for secondary and tertiary care, and patients are expected to contact the OOH-PCS by phone where triage is done prior to follow-up [16]. These differences in the organization of acute primary care may lead to differences in patient groups who consult the OOH-PCS [16] and thus also to differences in opioid prescribing.“

Methods

21. 

Page 4, line 113: I would think this is a retrospective cross sectional study

Author’s reply: This has been adjusted throughout the manuscript. 

22. 

Page 4, line 116 and line 121: Does iCAREdata cover 10% or 66% of the Flemish population?

Author’s reply: We apologize for this inconvenience. In this study, we only included data representative for 10% of the Flemish population. During the last years, iCAREdata grew extensively, leading to a population coverage of 66%. However, information is not available for these OOH-PCSs across the entire time horizon of this study. For this reason, the same subset of OOH-PCS was chosen to allow reliable comparison between the years. We revised the manuscript accordingly and removed the sentence in which we mention 66% of the Flemish population from the method section.

23. 

Page 6, line 146-148: I think it is unwise to exclude these data. The comparison between the regions seems like a secondary aim. You are excluding 2 million contacts per year...that's a lot. I would strongly encourage you to include these visits in your yearly descriptions and in the event you go back and perform a statistical analyses comparing the regions, you can then remove them. Trying to describe this very narrow subset of weekend contacts that result in an prescription is just not nearly as helpful.

Author’s reply: We followed the reviewers suggestion to repeat the analyses for the Netherlands including all contacts. This is now described in the method section of the manuscript. For comparison, the results excluding weekday and telephone contacts are included as supplementary material. Numbers in the abstract and in the results section were updated, as well as figures and supplementary material. The number of contacts in which an opioid was prescribed, slightly decreased. The changes over the years were similar to the results of the analysis excluding telephone and weekday contacts. Therefore the overall conclusion did not change. 

24. 

Page 6, line 150: ALL means need to have 95% CI

Author’s reply: we did not calculate means, but proportions and rates based on the entire datasets. We therefore think it is not meaningful to add 95% CIs. We updated terminology in the method section of the manuscript to clarify that we calculate rates and proportions and not means. 

Results

25. 

And please be consistent with the number of significant figures you are using.

Author’s reply: we checked the whole manuscript and changed the number of significance when inconsistent (1 in the text and 2 in tables). 

26. 

Page 6, line 171-172: Would re-write first sentence "Between 2015-2019, XXX (2.5%) OOH-contacts resulted in an opioid prescription (Figure 1, Table S1)"

Author’s reply: we changed this sentence as follows: “Between 2015 and 2019, approximately 2.5% of OOH contacts during the weekend resulted in an opioid prescription.”

27. 

Page 6-7, line 173-177: I am not following your % decrease calculation; going from 24  21% is an absolute reduction of 3%. 1-(21/24) = 12.5% relative reduction. If my math is incorrect please let me know

Author’s reply: We recalculated all relative reductions mentioned in the results section and changed where incorrect. 

28. 

Page 7, line 178-179: I would continue to present the mean (with 95% CI) values instead of switching to range.

Author’s reply: As explained in the reply to comment 24, we did not calculate means, but rates. For consistency in the presentation of results, we changed this paragraph as follows: “Prescription rates for strong opioids in OOH care decreased from 2.1 per 1000 contacts in 2015 to 1.3 in 2018 and increased to 1.8 per 1000 contacts in 2019 (14.3% relative decrease between 2015 and 2019). In the Netherlands, strong opioid prescribing increased from 9.7 to 13.3 per 1000 contacts between 2015 and 2017 (37.1 % increase) and slightly decreased thereafter to 12.3 per 1000 contacts in 2019.”

29. 

Page 7, line 188: Give % for each opioid

Author’s reply: This was now added for the year 2019. We would also like to refer to figure 2 in which the distribution of opioids is illustrated for each year and opioid group (strong vs weak).

Discussion

30. 

I would encourage you to include some discussion on how these results compare to other studies that describe opioid prescribing, particularly those that examine sources of opioid prescribing. Listed are references that I am familiar with as a US-based provider (Temporal Trends in Opioid Prescribing Practices in Children, Adolescents, and Younger Adults in the US From 2006 to 2018, JAMA Pediatr

. 2021 Oct 1;175(10):1043-1052. doi: 10.1001/jamapediatrics.2021.1832. Opioid Prescribing to US Children and Young Adults in 2019, Pediatrics

. 2021 Sep;148(3):e2021051539. doi: 10.1542/peds.2021-051539. Epub 2021 Aug 16. Variation in Adult Outpatient Opioid Prescription Dispensing by Age and Sex — United States, 2008–2018, MMWR Morb Mortal Wkly Rep. 2020 Mar 20;69(11):298-302. doi: 10.15585/mmwr.mm6911a5.).

Author’s reply: Comparison is difficult due to differences in setting (e.g. different patients contacting different settings, such as day time general practice, OOH services and emergency departments), health systems (gate keeper system, versus system with direct accessibility of secondary care) and the severity of the “opioid pandemic” (Western European countries versus US for example). Therefore, other than the comparison that was already mentioned (within the Netherlands), we chose to not add further comparison. 

31. 

Page 8, line 217-218: "Overall we did...nor in the Netherlands." Were you expecting to see a large increase? Was that your hypothesis? Seems odd to state this here, would delete.

Author’s reply: We indeed did not mention a hypothesis considering the direction of expected change in our manuscript. We thus deleted this sentence from the manuscript. 

32. 

Page 8, line 223-225: Are you saying that the overall trend of decreasing opioid prescribing is secondary to the increased knowledge of the dangers of opioids? It is confusing as this sentence just follows your statement that prescribing of strong opioids increased between 2015-2019 so it makes it seem like you are saying the attention being paid to prescribing has led to increased strong opioid prescribing.

Author’s reply: We meant to say that both the increase in strong opioid prescribing and the decrease in opioid prescribing from 2017 on, were consistent with opioid prescribing in the non acute primary care setting. Furthermore, the decrease is likely the result of increased attention for the risks of opioid prescribing from the end of 2017 on. We clarified this in the manuscript as follows: “The increase between 2015 and 2017 was driven by increased prescribing of strong opioids, while weak opioids prescribing remained stable. These changes (both the increase and the decrease thereafter) are consistent with the opioid prescribing trends in the Netherlands in the non-acute primary care setting [15]. The observed decrease in opioid prescribing is likely the result of increased attention since the end of 2017 for the steep rise in and risks of opioid prescribing.”

33. 

Page 9, line 254: Please provide a reference

Author’s reply: we added a reference to this sentence. 

34.

Page 9-10: Given the high prevalence of tramadol in your sample, I would include some discussion on the recent paper that showed increased mortality with tramadol (Association of Tramadol vs Codeine Prescription Dispensation With Mortality and Other Adverse Clinical Outcomes, JAMA. 2021 Oct 19;326(15):1504-1515. doi: 10.1001/jama.2021.15255.)

Author’s reply: The high tramadol prevalence is indeed notable. However, we did not include this reference in our manuscript, as tramadol users in the study suggested by the reviewer are compared with codeine users and in another study with NSAID users. In Flanders and the Netherlands, these patient groups differ greatly, where e.g. codeine is hardly used as pain medication, whereas tramadol is. Higher mortality for tramadol than codeine users could be expected. 

35. 

Limitations: IF you are going to continue to exclude the weekday contacts in the Netherlands, it needs to be listed in the limitations as this is excluding the majority of contacts from the Netherlands and significantly biases the results. As stated previously, would strongly encourage you to include these results and perform trend analysis.

Author’s reply: We followed the reviewer’s suggestion and included all contacts for the Netherlands (see reply to comment 23). 

Tables and Figures

36.

Try to use "." or "," as decimal points...you go back and forth in the supplemental material

Author’s reply: All decimal comma’s in supplementary table 2 were changes into “.”

37.

Figure 1: Year should be on the X-axis

Author’s reply: We amended the figure and put year on the X-axis.

---

## [Decision Letter · Decision Letter 1]

28 Feb 2022

Opioid prescribing in out-of-hours primary care in Flanders and the Netherlands: a retrospective cross-sectional study

PONE-D-21-33076R1

Dear Dr. Hek,

We’re pleased to inform you that your manuscript has been judged scientifically suitable for publication and will be formally accepted for publication once it meets all outstanding technical requirements.

Kind regards,

Vijayaprakash Suppiah, PhD

Academic Editor

PLOS ONE

Reviewers' comments:

Reviewer's Responses to Questions

**Comments to the Author**

1. If the authors have adequately addressed your comments raised in a previous round of review and you feel that this manuscript is now acceptable for publication, you may indicate that here to bypass the “Comments to the Author” section, enter your conflict of interest statement in the “Confidential to Editor” section, and submit your "Accept" recommendation.

Reviewer #2: All comments have been addressed

2. Is the manuscript technically sound, and do the data support the conclusions?

Reviewer #2: Yes

3. Has the statistical analysis been performed appropriately and rigorously? 

Reviewer #2: Yes

4. Have the authors made all data underlying the findings in their manuscript fully available?

Reviewer #2: Yes

5. Is the manuscript presented in an intelligible fashion and written in standard English?

Reviewer #2: Yes

6. Review Comments to the Author

Reviewer #2: Thank you for your thoughtful revisions. I commend you for your hard work. All comments have been addressed.

7. PLOS authors have the option to publish the peer review history of their article (what does this mean?). If published, this will include your full peer review and any attached files.

Reviewer #2: No

---

## [Editor Report · Acceptance letter]

30 Mar 2022

PONE-D-21-33076R1 

Opioid prescribing in out-of-hours primary care in Flanders and the Netherlands: a retrospective cross-sectional study 

Dear Dr. Hek:

I'm pleased to inform you that your manuscript has been deemed suitable for publication in PLOS ONE. Congratulations! Your manuscript is now with our production department. 

Kind regards, 

on behalf of

Dr. Vijayaprakash Suppiah 

Academic Editor

PLOS ONE